# POWER SCHEDULER: A BATCH SIZE AND TOKEN NUMBER AGNOSTIC LEARNING RATE SCHEDULER

## ABSTRACT

Finding the optimal learning rate for language model pretraining is a challenging task. This is not only because there is a complicated correlation between learning rate, batch size, number of training tokens, model size, and other hyperparameters but also because it is prohibitively expensive to perform a hyperparameter search for large language models with Billions or Trillions of parameters. Recent studies propose using small proxy models and small corpus to perform hyperparameter searches and transposing the optimal parameters to large models and large corpus. While the zero-shot transferability is theoretically and empirically proven for model size related hyperparameters, like depth and width, the zero-shot transfer from small corpus to large corpus is underexplored. In this paper, we study the correlation between optimal learning rate, batch size, and number of training tokens for the recently proposed WSD scheduler. After thousands of small experiments, we found a power-law relationship between variables and demonstrated its transferability across model sizes. Based on the observation, we propose a new learning rate scheduler, *Power scheduler*, that is agnostic about the number of training tokens and batch size. The experiment shows that combining the Power scheduler with Maximum Update Parameterization ($\mu$P) can consistently achieve impressive performance with one set of hyperparameters regardless of the number of training tokens, batch size, model size, and even model architecture. Our 3B dense and MoE models trained with the Power scheduler achieve comparable performance as state-of-the-art small language models.

## 1 INTRODUCTION

Learning rate is a critical hyperparameter for deep neural network training. In the context of Large Language Models (LLMs), the cosine learning rate scheduler is the most commonly used strategy. It has been shown to be effective across multiple state-of-the-art models, including Llama 3 (Dubey et al., 2024a), Gopher (Rae et al., 2021), etc. However, the cosine scheduler requires pre-defined training step counts to achieve the optimal loss. This results in two main drawbacks: 1) the intermediate training checkpoints are suboptimal, 2) continual training of an existing language model becomes complicated.

MiniCPM (Hu et al., 2024) proposes the Warmup-Stable-Decay (WSD) learning rate scheduler (illustrated in Figure 1) to address these issues. The WSD learning rate schedule is divided into three phases: 1) warmup phase, linearly increase the learning rate from 0 to peak; 2) stable phase, maintain the learning rate at peak value and training the model for most of the time; 3) decay phase, annealing the learning rate to 0 in a relatively short period. The main advantage of this schedule is that specifying the number of training steps in advance is not required. This is particularly convenient for large runs, as the decay can be applied at any time to observe model performance and decide whether to stop. It also allows for continual learning by default, as training can be resumed from a stable phase checkpoint. Moreover, the data mixture can be changed during the decay phase to increase the ratio of high-quality data. This data curriculum is shown to be effective in several recent language models (Team et al., 2023; Hu et al., 2024; Dubey et al., 2024a; Shen et al., 2024).

However, is WSD really agnostic to token count? In our experiments, we found that although the WSD scheduler could, in theory, continue in the stable phase forever, the optimal learning rates are different for different amounts of training tokens. In other words, the optimal learning rate scheduler

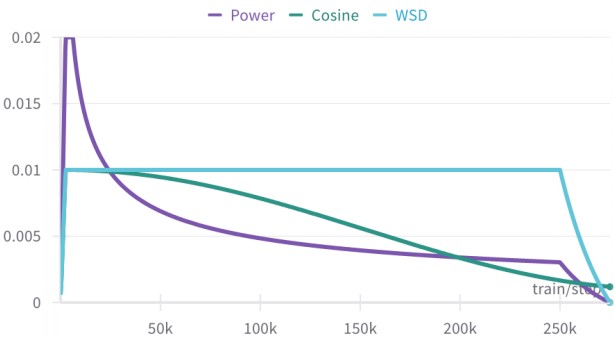

Figure 1: Illustration of learning rate curves for Cosine, WSD, and our Power schedulers.

is tied to the number of training tokens. Thus, the WSD scheduler still faces the same issue as the cosine scheduler: the intermediate and continual training checkpoints are suboptimal if the number of tokens is too different from the original plan.

Furthermore, deciding the optimal learning rate is still challenging for large-scaling pretraining. As our experiments will show, there is a complicated correlation between hyperparameters, including learning rate, model size, batch size, and number of training steps. The size and training cost of modern LLMs make it impossible to do a hyperparameter search on the target model size and training flops. Researchers have proposed using small proxy models to run hyperparameter searches and predict the optimal hyperparameters of large models from search results (Dey et al., 2023; Hu et al., 2024; Yang et al., 2022). Among these methods, $\mu$Transfer (Yang et al., 2022; 2023) is proposed to facilitate zero-shot hyperparameter transfer between different model sizes. $\mu$Transfer has been successfully applied to several language models, including Cerebras-GPT (Dey et al., 2023), miniCPM (Hu et al., 2024), and AFM (Gunter et al., 2024).

In this paper, we first combine the WSD scheduler and $\mu$Transfer to study the learning rate transfer between proxy and large models. Our extensive experiments show that $\mu$Transfer does not provide direct zero-shot learning rate transferability across the numbers of tokens and batch sizes for the WSD optimizer. Instead, the optimal learning rate $\eta_{opt}$ satisfies a power-law relation with respect to batch size $\beta$ and number of tokens $T$:

$$\eta_{opt} = \beta \cdot aT^b \tag{1}$$

where $a$ and $b$ are power-low coefficients. Furthermore, our experiment confirms the zero-shot transferability across model sizes, different model sizes share very similar coefficients. Inspired by this observation, we propose a new learning rate scheduler, PowerLR, that is agnostic to batch size and token number. It allows direct transfer of the optimal learning rate scheduling across batch size, token numbers, and model size. Thus, the expensive pretraining runs can be trained without specifying the number of training tokens, enabling early stop and continual pretraining without sacrificing convergence.

## 2 BACKGROUND

### 2.1 MAXIMUM UPDATE PARAMETRIZATION ($\mu$P)

Maximal Update Parameterization ($\mu$P) (Yang & Hu, 2020; Yang et al., 2022; 2023) controls initialization, layer-wise learning rates, and activation magnitudes to ensure analytically stable training, independent of a model's width and depth. In addition to improving training stability, $\mu$P improves the transferability of training hyperparameters from small proxy models to large models, a technique called $\mu$Transfer. The hyperparameter transferability of $\mu$P is theoretically justified for width (Yang et al., 2022) and depth (Yang et al., 2023). Previous work also show empirical evidence for transferability across batch size, training steps, and training sequence length.

In this paper, we follow the $\mu$P config used in CerebrasGPT (Dey et al., 2023) to study the transferability of batch size and learning rate across different numbers of training tokens and model sizes. Table 1 lists the $\mu$P changes we applied to model initialization, learning rate, and multipliers.

Table 1: List of changes applied when using $\mu$P. $m_{\text{width}}$ is width multiplier, defined as $d_m/d_{\text{base}}$, where $d_{\text{base}}$ is the embedding width, $d_m$ is the target model size.

| Name | Function |
|------|----------|
| Embedding multiplier | Multiply the embedding output with $m_{\text{emb}}$ |
| Residual Multiplier | Multiply the output of each attention and MLP layer with $m_{\text{res}}$ before adding to residual connection |
| Initialization std | Initialize internal weight matrices (excluding input and output embedding) with standard deviation $\text{init}_{\text{std}}/\sqrt{m_{\text{width}}}$ |
| Learning rate scaling | Set learning rate of internal weight matrices to $\eta/m_{\text{width}}$ |
| Attention logit scaling | Divide attention logits by $d_{\text{head}}$ |

## 2.2 WARMUP-STABLE-DECAY (WSD) SCHEDULER

MiniCPM (Hu et al., 2024) proposes the WSD learning rate scheduler to divide pretraining into three stages: the warmup stage, the stable training stage, and the remaining decay stage. The WSD scheduler is defined as:

$$\text{WSD}(n) = \begin{cases} \frac{n}{N_{\text{warmup}}} \cdot \eta & \text{if } n < N_{\text{warmup}} \\ \eta & \text{if } N_{\text{warmup}} < n \leq N - N_{\text{decay}} \\ f(n, N, N_{\text{decay}}) \cdot \eta & \text{if } n > N - N_{\text{decay}} \end{cases}, \tag{2}$$

where $n$ is the current number of steps, $\eta$ is the stable learning rate, $N$ is the total number of steps, $N_{\text{warmup}}$ is the number of warmup steps, $N_{\text{decay}}$ is the number of decay steps, $f(n, N, N_{\text{decay}})$ is the learning rate decay function. The main advantage of WSD scheduler over other schedulers (e.g. cosine and linear) is that specifying the number of training steps in advance is not required. Since the learning rate in the stable training stage does not depend on the number of training steps, the WSD scheduler does not need to specify the number of training steps in advance. This is particularly convenient for large runs, as the cooldown can be initiated at any time to observe model behavior and decide whether to stop (Hägele et al., 2024). It also facilitates extended pre-training, which can be applied to the last checkpoint in the stable training phase. Moreover, the data mixture can be changed during the cooldown phase to incorporate more high-quality data toward the end of the training. This data mixture strategy has been proven effective in many recent language models (Hu et al., 2024; Dubey et al., 2024a; Team et al., 2024).

## 3 SEARCH OPTIMAL LEARNING RATE FOR WSD SCHEDULER WITH $\mu$P

In this section, we focus on finding the optimal learning rate $\eta$ for a small proxy model. Due to our use of $\mu$P, we expect the optimal learning rate to be invariant across different model sizes, batch sizes, and training steps. To verify this, we conducted extensive experiments to find the optimal learning rate for different model sizes and numbers of training tokens. Table 2 shows our overall hyperparameter search configurations. We use the WSD scheduler with 10% tokens in the decay phase following previous works (Hu et al., 2024; Hägele et al., 2024). All models in this section and the next section are trained on the RedPajama (Computer, 2023) corpus and tested on a holdout test set.

## 3.1 DOES OPTIMAL LEARNING RATE TRANSFER?

We conduct two controlled experiments to verify the optimal learning rate's transferability. First, we fix the batch size $\beta = 128$ and swept across all combinations of learning rates $\eta$ and numbers of training tokens $T$. Figure 2(a) shows that the optimal learning rate, $\eta_{\text{opt}}$, does not transfer across a wide range of $T$. $\eta_{\text{opt}}$ tends to decrease with respect to the number of training tokens. For example, $\eta_{\text{opt}}$ is 0.0128 for 2B training tokens and 0.0008 for 256B. Second, we fix the number of training tokens $T$ to 128 billion tokens and swept across all combinations of learning rates $\eta$ and batch sizes.

Table 2: Hyperparameter search for finding the optimal learning rate and batch size combination across different numbers of training tokens.

| Model size | 12M | 36M | 121M |
|---|---|---|---|
| *Fixed hyperparameters* | | | |
| Attention head size | 64 | 64 | 64 |
| Number of layers | 32 | 32 | 32 |
| Number of attention heads | 2 | 4 | 8 |
| Embedding size | 128 | 256 | 512 |
| MLP hidden size | 320 | 640 | 1280 |
| Initialization std | 0.02 | 0.02 | 0.02 |
| Sequence Length | 4096 | 4096 | 4096 |
| Warmup tokens | 1B | 1B | 1B |
| $m_{\text{width}}$ | 0.5 | 1 | 2 |
| $m_{\text{emb}}$ | 1 | 1 | 1 |
| $m_{\text{residual}}$ | 1 | 1 | 1 |
| *Variable hyperparameters* | | | |
| Training tokens $T$ | 2B, 4B, 8B, 16B, 32B, 64B, 128B, 256B | | |
| Batch size $\beta$ | 16, 32, 64, 128, 256, 512 | | |
| Learning rate $\eta$ | 0.0002, 0.0004, 0.0008, 0.0016, 0.0032, 0.0064, 0.0128, 0.0256 | | |

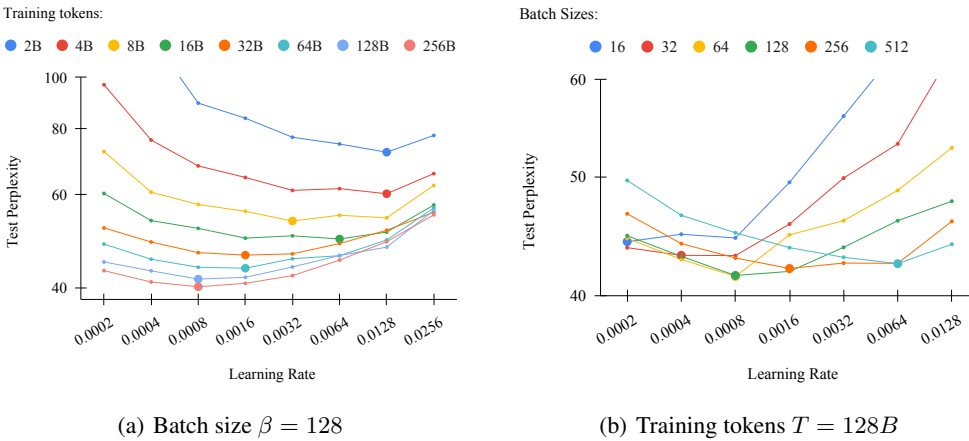

(a) Batch size $\beta = 128$        (b) Training tokens $T = 128B$

Figure 2: **Left:** Learning Rate v.s. Test Perplexity for different numbers of training tokens. The optimal learning rate decreases with respect to the number of training tokens. **Right:** Learning Rate v.s. Test Perplexity for different batch sizes. The optimal learning rate increases with respect to the batch size.

Figure 2(b) shows that the optimal learning rates do not transfer across a wide range of $\beta$, but tends to increase with respect to the batch size. The optimal learning rate $\eta_{\text{opt}}$ is 0.0002 for batch size 16 and 0.0064 for batch size 512.

## 3.2 WHAT IS THE RELATIONSHIP BETWEEN $\eta_{\text{opt}}$, $\beta$ AND $T$?

To better understand the relationship between $\eta_{\text{opt}}$, $\beta$, and $T$, we swept across all possible combinations of these three variables. Figure 3 shows the optimal learning rate for each batch size and training token combination. Like our previous observation, the optimal learning rate consistently decreases with respect to the number of training tokens across different batch sizes. Furthermore, we also notice that the ratio between the optimal learning rate $\eta_{\text{opt}}$ and batch size $\beta$ is relatively stable for each number of training tokens. Based on this observation, we make our first hypothesis:

**Hypothesis 1** *The optimal learning rate $\eta_{\text{opt}}$ for the WSD scheduler and a given pair of $(T, \beta)$ is proportional to the training batch size $\beta$.*

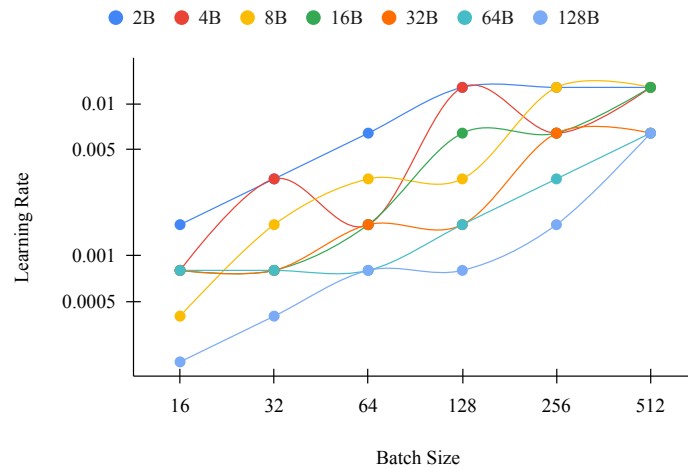

Figure 3: Batch Size v.s. Learning Rate for different numbers of training tokens. The optimal learning rate increases with respect to the batch size.

Thus, we define $\gamma$ as the ratio between $\eta_{\text{opt}}$ and $\beta$:

$$\gamma = \frac{\eta_{\text{opt}}}{\beta} \tag{3}$$

To verify Hypothesis 1, we conducted an extensive hyperparameter search for three model sizes: 12M, 36M, and 121M. After finding the optimal learning rate $\eta_{\text{opt}}$ for every combination of $(T, \beta, \text{model size})$, we only keep the three best batch sizes to focus on the optimal scenario. The $\gamma$ of the three best batch sizes for each $T$ and model size are plotted in in Figure 4(a), Figure 4(b), and Figure 4(c). These results show that, given a fixed number of training tokens $T$, $\gamma$ falls in a relatively small region.

Furthermore, we notice that $\gamma$ approximately follows a power-law relation with respect to the number of training tokens. Thus, we make a second hypothesis:

**Hypothesis 2** *The Learning rate to batch size ratio $\gamma$ has a power-law correlation with $T$:*

$$\gamma = aT^b \tag{4}$$

*when using μP, the correlation can be transferred across model sizes.*

To verify the Hypothesis 2, we compute the average $\gamma$ of the best batch sizes to estimate the real $\gamma$ for each number of tokens $\beta$. Figure 4(d) shows that all three model sizes share a similar power-law correlation. After aggregating the results from three sizes, we get the following correlation:

$$\gamma = 4.6T^{-0.51} \tag{5}$$

where $x$ is the number of training tokens.

In other words, the relation between optimal learning rate $\eta_{\text{opt}}$ for the WSD scheduler, batch size $\beta$, and the number of training tokens $T$ can be approximately modeled with the following equations:

$$\eta_{\text{opt}} = \beta \cdot aT^b \tag{6}$$

Equation 6 provides an easy way to predict the optimal learning rate given the number of training tokens and training batch size for the WSD scheduler. As in prior work, $a$ and $b$ can be easily estimated through hyperparameter search on a small proxy model.

However, this relationship between the number of training tokens and the optimal learning rate constrains the number of training steps to be specified before training. We also noticed that the equation tends to provide a small learning rate for a large-scale training corpus. For example, given a 10 trillion token corpus and a batch size 1024, the predicted optimal learning rate is 0.0011. If we

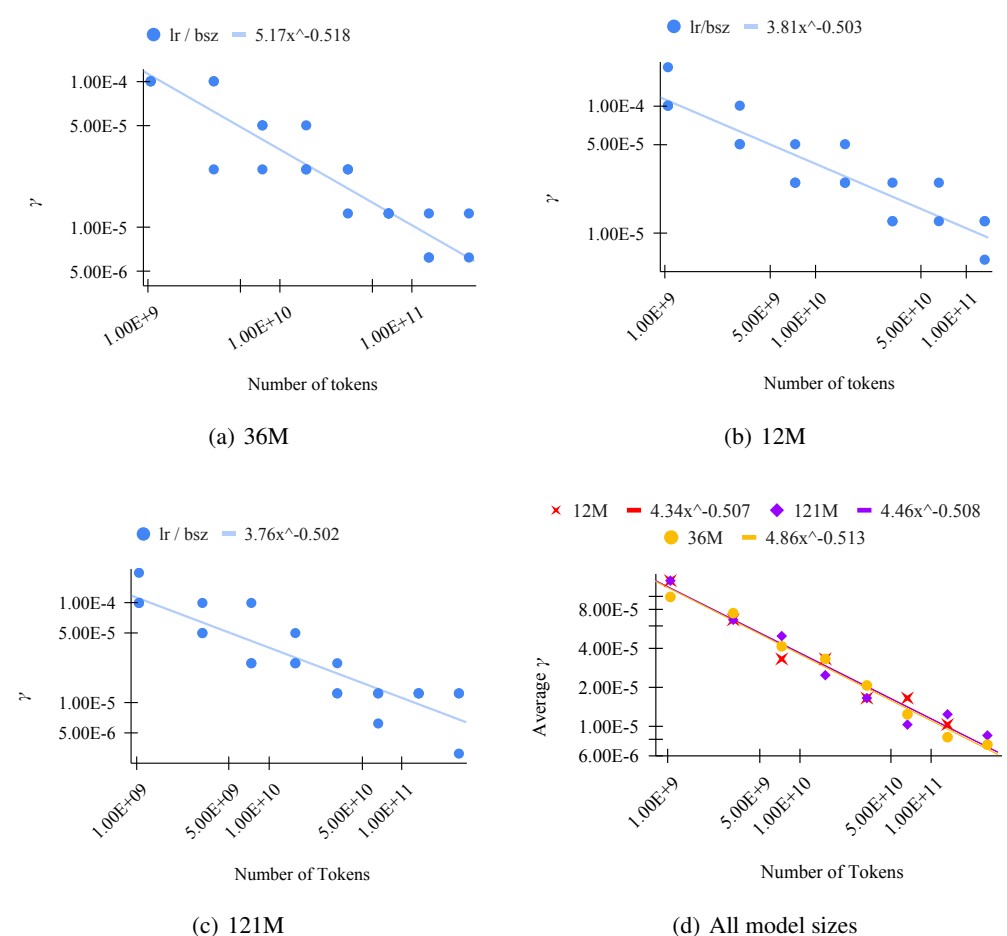

Figure 4: **a, b, c:** The $\gamma$ of best three batch sizes v.s. The number of training tokens. (Dots could overlap.) **d:** The average $\gamma$ of the three best batch sizes v.s. The number of training tokens. A power-law function can model the correlation between $\gamma$ and $T$.

consider that $\mu$P will divide the learning rate by a factor of $m_{\text{width}} = d_{\text{model}}/d_{\text{base}}$ for matrices inside the model, the actual learning rate will be very small for large models. While a small learning rate is good for final convergence, it is also likely to cause insufficient exploration at the beginning of training. To solve these issues, we propose a novel power learning rate scheduler in the next section.

## 4  POWER SCHEDULER

Inspired by previous observations, we propose a new power learning rate based on the observation from the previous section:

$$\eta_{\text{power}}(n) = \min\left(\eta_{\text{max}}, \beta \cdot an^b\right) \tag{7}$$

where $\beta$ is the batch size, $n$ is the number of tokens already trained, $a$ is the amplitude of the learning rate, $b$ is a power-law exponent for decaying the learning with respect to the number of trained tokens, and $\eta_{\text{max}}$ is the learning rate upper bound that rejects very large learning. Like the constant learning rate, the power learning rate also does not require specifying the number of training steps or total training tokens beforehand, since the learning rate only depends on the current number of training tokens.

Like the WSD scheduler, we can combine the power learning rate with warmup and decay to get the final Power scheduler:

$$\text{Power}(n) = \begin{cases} \frac{n}{N_{\text{warmup}}} \cdot \eta_{\text{power}}(N_{\text{warmup}}) & \text{if } n < N_{\text{warmup}} \\ \eta_{\text{power}}(n) & \text{if } N_{\text{warmup}} < n \leq N - N_{\text{decay}} \\ f(n, N, N_{\text{decay}}) \cdot \eta_{\text{power}}(N - N_{\text{decay}}) & \text{if } n > N - N_{\text{decay}} \end{cases} \quad (8)$$

Table 3: Hyperparameter search config for Power scheduler. The search range for $a$ and $b$ is decided based on our observation in Section 3.

| Model size | 36M |
|---|---|
| *Fixed hyperparameters* | |
| Attention head size | 64 |
| Number of layers | 32 |
| Number of attention heads | 4 |
| Embedding size | 256 |
| MLP hidden size | 640 |
| Initialization std | 0.1 |
| Sequence Length | 4096 |
| Warmup tokens | 1B |
| $m_{\text{width}}$ | 1 |
| $m_{\text{emb}}$ | 12 |
| $m_{\text{residual}}$ | 0.26 |
| Max learning rate $\eta_{\text{max}}$ | 0.02 |
| *Variable hyperparameters* | |
| Batch size $\beta$ | 32, 64, 128, 256, 512 |
| Training tokens $T$ | 2B, 4B, 8B, 16B, 32B, 64B, 128B |
| $a$ | $[3, 5]$ |
| $b$ | $[-0.6, -0.4]$ |

The Power scheduler requires three separate hyperparameters $(\eta_{\text{max}}, a, b)$, instead of only one hyperparameter required by the cosine and WSD schedulers. However, we expect these hyperparameters to transfer across different model sizes, number of training tokens, and batch sizes. Additionally, since Figure 1 shows that the $\eta_{\text{max}}$ has a very limited impact on the overall learning rate curve, we can simply set it to a large enough value. In this paper, we set $\eta_{\text{max}}$ to $0.02$. Then, we conduct another hyperparameter search for $a$ and $b$ to find the optimal learning rate hyperparameter. Table 3 shows the configuration used in the Power scheduler hyperparameter search.

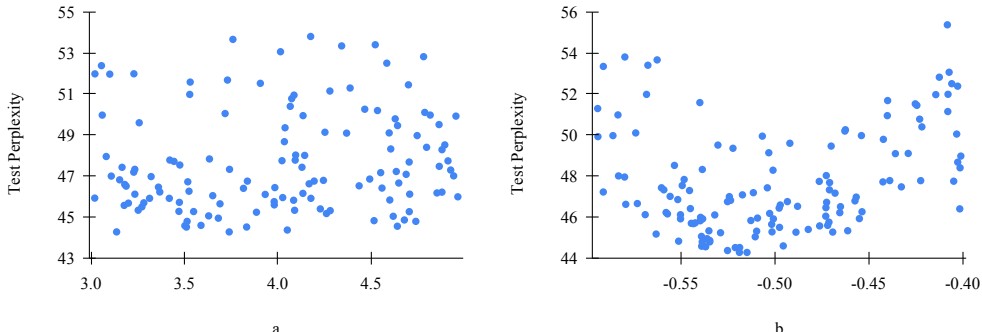

Figure 5: Hyperparameter search results for 32B training tokens. **Left:** Test perplexity vs $a$. The test perplexity is not sensitive to different choices of $a$ within the range of $[3, 5]$. **Right:** Test perplexity vs $b$. The test perplexity is more sensitive to the choices of $b$. The optimal $b$ is between $-0.52$ and $-0.51$.

Figure 5 shows the hyperparameter search results for 32 billion training tokens. We observe that the test perplexity is not sensitive to different choices of $a$ within our search range of $[3, 5]$. In contrast,

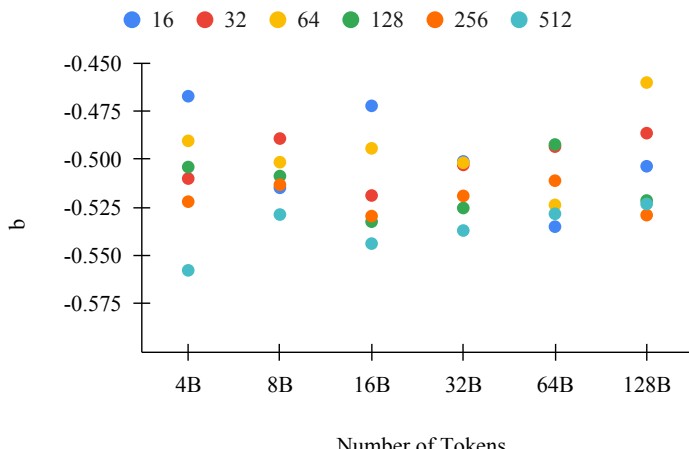

Figure 6: Optimal $a$ vs Number of training tokens. The optimal $a$ is consistent across different numbers of training tokens.

the test perplexity is more sensitive to the choices of $b$. The optimal $b$ is between $-0.52$ and $-0.51$. A similar observation has been made across different numbers of training tokens. Figure 6 shows that the optimal $b$ for different $\beta$ and $T$ consistently falls within the range of $[-0.49, -0.53]$ with few exceptions. Combining this and previous observations, we select $a = 4$ and $b = -0.51$ for the rest of this paper.

## 5 PRE-TRAINING EXPERIMENTS

This section compares the Power scheduler with the WSD and Cosine scheduler across different scenarios. In the first part, we conduct a controlled experiment to train 1B transformer models and 1B mixture-of-experts (MoE) models with different learning rate schedulers to show that the Power scheduler is comparable to or better than other schedulers. In the second part, we take a more realistic setting, training a 3B transformer language model and a 3B MoE language model with high-quality data in the decay phase to compare with strong open-source language models. For all the Power scheduler experiments in this section, we will use the hyperparameters from Section 4, $a = 4$, $b = -0.51$, and $\eta_{\max} = 0.02$.

### 5.1 1B CONTROLLED EXPERIMENT

We train a series of 1B parameter dense and Mixture-of-Experts (MoE) transformer models using WSD, cosine, and Power schedulers. All models are trained with 1T tokens. We use the optimal hyperparameters proposed in MiniCPM (Hu et al., 2024) and adapt them to our 1B setting with $\mu$Transfer. All models are trained with batch size 1024. For WSD and cosine scheduler, we set $\eta = 0.01$, following the optimal learning rate from MiniCPM. For the Power and WSD scheduler, we exponentially decay the learning rate to 0 for the last 100B tokens. The MoE models are implemented with ScatterMoE (Tan et al., 2024). More model details can be found in Table 4.

We evaluate all 1B models on language model tasks and multiple-choice tasks from LM evaluation Harness (Gao et al., 2024). The multiple-choice tasks include grade-school science questions (ARC, Clark et al. (2018)), yes/no questions (BoolQ, Clark et al. (2019)), common sense reasoning (Hellaswag, Zellers et al. (2019)), open book question answering (OpenBookQA, Mihaylov et al. (2018)), physical questions (PIQA, Bisk et al. (2020)), and Winograd schema task (Winogrande, Sakaguchi et al. (2021)). Table 5 shows the performance. The Power scheduler provides consistently better or comparable performance for both language modeling and downstream tasks. Surprisingly, even though we only performed a hyperparameter search on a small dense model, the hyperparameters still performed well on the MoE model.

Table 4: Hyperparameter for 1B and 3B models.

| Model | 1B Dense | 1B MoE | PowerLM-3B | PowerMoE-3B |
|---|---|---|---|---|
| Embedding size | 1536 | 1024 | 2304 | 1536 |
| Number of layers | 40 | 24 | 40 | 32 |
| Attention head size | 64 | 64 | 64 | 64 |
| Number of attention heads | 24 | 16 | 36 | 24 |
| Number of KV heads | 24 | 8 | 36 | 8 |
| MLP hidden size | 4096 | 512 | 9216 | 512 |
| MLP activation | swiglu | swiglu | swiglu | swiglu |
| Number of Experts | – | 32 | – | 40 |
| MoE TopK | – | 8 | – | 8 |
| Initialization std | 0.1 | 0.1 | 0.1 | 0.1 |
| Sequence Length | 4096 | 4096 | 4096 | 4096 |
| $m_{\text{width}}$ | 6 | 4 | 8 | 6 |
| $m_{\text{emb}}$ | 12 | 12 | 12 | 12 |
| $m_{\text{residual}}$ | 0.22 | 0.28 | 0.22 | 0.22 |
| #Parameters | 1.2B | 1.3B | 3.5B | 3.3B |
| #Active Parameters | 1.2B | 377M | 3.5B | 800M |
| #Training tokens | 1T | 1.1T | 1.25T | 3T |

Table 5: Language Modeling and Zero-shot performance of 1B models. $acc_n$ means accuracy with average log probability.

| Task | | Wiki | ARC | BoolQ | Hellaswag | OBQA | PIQA | WinoGrande | **Average** |
|---|---|---|---|---|---|---|---|---|---|
| Metric | | $ppl$ | $acc_n$ | $acc$ | $acc_n$ | $acc_n$ | $acc_n$ | $acc$ | |
| Dense | Cosine | 14.5 | **44.6** | 63.5 | 63.6 | 37.6 | 75.3 | 61.3 | 57.7 |
| | WSD | 13.9 | 43.0 | 62.8 | **64.8** | **38.0** | 74.5 | **62.1** | 57.6 |
| | Power | **13.8** | 44.3 | **65.6** | 64.6 | 37.0 | **76.0** | 61.8 | **58.2** |
| MoE | Cosine | 15.7 | 40.6 | **63.6** | 59.7 | 35.2 | 74.0 | 57.2 | 55.1 |
| | WSD | 14.8 | 43.9 | 59.7 | 61.0 | 34.4 | **76.0** | **59.3** | 55.7 |
| | Power | **14.6** | **44.0** | 60.4 | **61.6** | **36.0** | 74.8 | 58.3 | **55.9** |

## 5.2 3B REALISTIC EXPERIMENT

To compare performance against strong open-source language models, we pretrain two 3B language models: 1) PowerLM-3B, a 3B dense language model, and 2) PowerMoE-3B, a 3B MoE language model. We pretrain these two models using the two-stage training schema in Hu et al. (2024) and Power scheduler. Stage 1 linearly warms up the learning rate and then applies the power decay. The training corpus is a mix of large-scale, medium-quality open-source datasets with permissive licenses. PowerLM-3B is trained on 1T tokens, and PowerMoE-3B on 2.5T tokens. Stage 2 exponentially decays the learning rate to zero. The training corpus is a mix of stage 1 data and a small amount of high-quality open-source/synthetic corpora with permissive licenses. PowerLM-3B is trained on 250B tokens, and PowerMoE-3B on 500B tokens. The training batch size is 1024.

Table 6: Zero-shot performance on multiple-choice tasks.

| Task | ARC | BoolQ | Hellaswag | OBQA | PIQA | WinoGrande | **Average** |
|---|---|---|---|---|---|---|---|
| Metric | $acc_n$ | $acc$ | $acc_n$ | $acc_n$ | $acc_n$ | $acc$ | |
| Qwen1.5-4B | 50.6 | **77.7** | 71.5 | 39.8 | 77.0 | 64.4 | 63.5 |
| Gemma2-2B | **65.0** | 72.8 | 73.0 | 41.4 | 79.2 | 68.9 | 66.7 |
| PowerLM-3B | 60.5 | 72.0 | **74.6** | **43.6** | **79.9** | **70.0** | **66.8** |
| Qwen1.5-1.8B | 47.1 | 66.3 | 60.9 | 34.2 | 74.0 | 60.8 | 57.2 |
| Qwen2-1.5B | 48.1 | **72.5** | 65.4 | 36.4 | 75.3 | **66.2** | 60.7 |
| SmolLM-1.7B | **59.9** | 66.1 | 65.6 | **42.0** | 75.7 | 60.5 | 61.7 |
| PowerMoE-3B | 58.1 | 65.0 | **71.5** | 41.0 | **79.1** | 65.0 | **63.3** |

Table 7: Performance on Math, Coding, and MMLU. The MATH 5-shot, exact matching setting is from Hendrycks et al. (2021), and the 4-shot, sympy matching setting is from Lewkowycz et al. (2022).

| Task | GSM8k | | MATH | | HumanEval | MBPP | MMLU |
|------|-------|------|------|------|-----------|------|------|
| Prompt | 5-shot | 8-shot,cot | 5-shot | 4-shot | 0-shot | 0-shot | 5-shot |
| Metric | *exact* | *exact* | *exact* | *sympy* | *pass@1* | *pass@1* | *acc* |
| Qwen1.5-4B | **53.1** | **54.8** | 9.76 | 12.8 | 7.3 | 30.6 | **55.2** |
| Gemma2-2B | 23.9 | 25.5 | 9.80 | **15.4** | 19.5 | 27.8 | 53.0 |
| PowerLM-3B | 34.9 | 49.7 | **9.90** | 15.2 | **26.8** | **33.6** | 49.2 |
| SmolLM-1.7B | 6.6 | 6.8 | 4.86 | 3.0 | **20.7** | 30.6 | 28.8 |
| Qwen1.5-1.8B | 34.7 | 37.3 | 5.66 | 9.6 | 4.30 | 21.6 | 45.6 |
| Qwen2-1.5B | **58.3** | **58.6** | 4.38 | **23.3** | 16.5 | 31.0 | **55.9** |
| PowerMoE-3B | 25.9 | 38.4 | **9.26** | 14.8 | 20.1 | **32.4** | 42.8 |

Table 6 shows the multi-choices performance of our model and state-of-the-art models. Table 7 shows MMLU and generative performance in the math and code domain. The results show that, despite being trained with relatively fewer tokens, our PowerLM-3B still achieves comparable performance with state-of-the-art 2B to 4B language models. Furthermore, our PowerMoE-3B uses only 800M active parameters but performs similarly to state-of-the-art 1B to 2B dense models.

## 6 RELATED WORKS

### 6.1 HYPERPARAMETER SEARCH FOR LLMS

Hyperparameters significantly impact a model's performance. However, directly adjusting hyperparameters for LLMs is not feasible. Tensor Program (Yang et al., 2022; 2023) proposes a framework to stabilize the hyper-parameters for models with different scales. In this framework, one could search for optimal hyperparameters at a smaller scale and zero-shot transfer the results to a much larger scale. CerebrasGPT (Dey et al., 2023) and MiniCPM (Hu et al., 2024) applied this method in their pretraining. Beyond the Tensor Program, researchers also leverage scaling law (Kaplan et al., 2020) to help decide the optimal hyperparameters (Dubey et al., 2024a; Xie et al., 2024). They first build a prediction function for the validation loss or accuracy from model hyperparameters (e.g., model size, training tokens, etc.), then leverage the function to predict the optimal hyperparameters.

### 6.2 LEARNING RATE SCHEDULER FOR LLMS

In the context of LLMs, the cosine scheduler is the most commonly used learning rate scheduler. Majority of LLMs, including GPT (Radford, 2018; Radford et al., 2019; Brown, 2020) and large models like Gopher (Rae et al., 2021), PaLM2 (Anil et al., 2023) or LLaMA (Touvron et al., 2023a;b; Dubey et al., 2024b), all use cosine scheduler as the de-facto standard schedule. However, recent LLMs have started to use multi-step learning rate schedulers to better accommodate multi-phase training data. Deepseek LLM (Bi et al., 2024) introduced a scheduler that decrease the learning rate twice during training, creating a 3-level step-like learning rate curve. MiniCPM (Hu et al., 2024) introduced a Warmup-Stable-Decay style learning rate that starts learning rate decay at the last 10% steps.

## 7 CONCLUSION

In this paper, we systematically study the relationship between optimal learning rate, batch size, and number of training tokens. We observed a power-law relation between these variables while using the WSD learning rate scheduler. Inspired by this observation, we propose a new learning rate scheduler, the Power scheduler, that is invariant with respect to the number of tokens and batch size. The experiment shows that combining the Power scheduler with Maximum Update Parameterization ($\mu$P) can consistently achieve impressive performance with one set of hyperparameters regardless of the number of training tokens, batch size, model size, and even model architecture.

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
