# OpenReview forum: "Power Scheduler: A Batch Size and Token Number Agnostic Learning Rate Scheduler"
_ICLR.cc/2025/Conference — ICLR 2025 Conference Withdrawn Submission_

### Official Review · Reviewer_axY6 · 2024-10-27

**Soundness:** 2
**Presentation:** 3
**Contribution:** 3
**Rating:** 6
**Confidence:** 3

**Summary:**

Power Scheduler is designed to be independent of the batch size and number of training tokens. The motivation for this work is the inefficiencies in current schedulers like the Warmup-Stable-Decay (WSD) scheduler, which still ties the optimal learning rate to the number of training tokens, making continual training and early stopping suboptimal for large language models (LLMs). By identifying a power-law relationship between optimal learning rates, batch size, and the number of training tokens, the authors introduce a new scheduling method that generalizes across model sizes, batch sizes, and token counts. The results show that this approach improves model performance while maintaining flexibility in training large models, including both dense and Mixture of Experts (MoE) models.

**Strengths:**

The Power Scheduler addresses a critical gap in learning rate scheduling for LLMs by eliminating the need to specify batch size or token counts in advance. This is a significant improvement over traditional approaches like the cosine and WSD schedulers, which require predefined training steps and can be suboptimal in continual learning scenarios. The paper introduces a well-motivated, novel concept of using a power-law relation to generalize the learning rate across different training setups, showing strong theoretical justification and empirical validation. The experimental results are demonstrating that the Power Scheduler achieves comparable or superior performance on both dense and MoE models. Its flexibility across model sizes and architectures makes it highly relevant for scaling deep learning workloads efficiently.

Another strength of the paper is its comprehensive experimental setup, which tests the Power Scheduler against state-of-the-art models in controlled environments and realistic training scenarios. The authors thoroughly explore hyperparameter search and transferability, particularly in the context of large-scale pretraining with billions of parameters. the integration with Maximum Update Parameterization (*µP*) shows that the scheduler can maintain stable training dynamics across various model sizes, a critical feature for improving the transferability of hyperparameters from small proxy models to large-scale models.

**Weaknesses:**

The major limitation of the paper is only 12M 36M 121M models are trained which significantly weakens the liability of results. For example, the ones in [1] involves multiple scales that are larger than ~100M models.

The paper could benefit from a more in-depth discussion on the computational costs of this search process, particularly for very large models where even proxy model training can be expensive. Offering practical guidelines for efficiently setting or approximating these hyperparameters without extensive searches would improve the usability of the method. In particular roughly how many FLOPs required for hyper-parameter sweeps, especially for the kind of MoE architecture employed in the paper.

The writing is generally good, but the large table of hyper-parameters disrupts the flow. Tab 3 and 4 are better to be in Appendix.

While the experiments demonstrate strong performance on language modeling some downstream tasks, including more challenging or varied domains, such as reasoning-based tasks or multi-modal data, would better validate the generality of the Power Scheduler. This can be true when we use post-training data in the decay phase, as did in MiniCPM.

Some related works are missing: Scaling Optimal LR Across Token Horizons. This paper show that learning rate should be tuned down if we increase token size.

[1] Small-scale proxies for large-scale Transformer training instabilities

**Questions:**

Why are the curves in fig. 3 very non-monotonic?

In fig. 2(a) training longer requires a larger LR in general, but the other paper found the opposite is true in a larger scale, what cause the difference? [1]

[1] Scaling Optimal LR Across Token Horizons

---

### Official Review · Reviewer_2PGW · 2024-10-30

**Soundness:** 3
**Presentation:** 3
**Contribution:** 3
**Rating:** 5
**Confidence:** 3

**Summary:**

The paper introduces the Power Scheduler, a novel learning rate scheduler for large language model (LLM) training that is invariant to batch size and the number of training tokens. The authors first explore the limitations of traditional schedulers, particularly the cosine and WSD schedulers, which are dependent on fixed training steps, leading to suboptimal checkpoints for models trained with varying token counts.

The key contributions are as follows:
1. Through theoretical and empirical studies, the paper demonstrates that the optimal learning rate follows a power-law relationship with batch size and token count, enabling a consistent learning rate setting across different model sizes and token volumes.

2. The scheduler is evaluated on various language models, including dense and MoE models, and shows that it performs comparably to or better than existing schedulers, even with different model sizes and architectures. The experiments further validate that Power Scheduler can transfer hyperparameters across proxy models to large-scale models, offering a cost-effective solution for extensive LLM training.

**Strengths:**

1. The paper is well-written and easy to understand.

2. The research topic on learning rate scheduler is timely and important for the LLM community.

3. Theoretically, the proposed method is built upon the mu-P framework and thus has a good foundation.

4. Empirically, the effectiveness of the proposed method is verified on up to 3B dense and MoE models, which is comprehensive and convincing.

**Weaknesses:**

1. Lack of theoretical justifications and ablation studies for the design choice in Eq. (7). To solve the issue that Eq. (6) tends to provide a small learning rate for a large-scale training corpus, the authors propose a specific design choice in Eq. (7). However, it is unclear how this design choice aligns with the previous result built upon mu-P theory and whether there are better choices.

2. The improvement over WSD seems marginal. As shown in Table 5, measured by PPL, the gap between Cosine and WSD is obvious. However, the proposed method performs similarly to WSD, +0.1 and +0.2 for dense and MoE respectively.

I am open to raising my score if the authors address the above Weaknesses.

**Questions:**

1. Besides model size and architectures, can the numbers a=4 and b=0.51 transfer to other optimizers or tokenizers? Do we need to search them again on the corresponding proxy models?

2. How to interpret the results in Table 6 and Table 7? If I understand correctly, there exist many differences between the PowerLM and the compared models, not solely the Power Scheduler. How could we attribute the performance gain to the proposed method?

---

### Official Review · Reviewer_ZEew · 2024-10-31

**Soundness:** 2
**Presentation:** 3
**Contribution:** 2
**Rating:** 3
**Confidence:** 4

**Summary:**

The paper addresses the challenge of optimizing the learning rate for language model pretraining, complicated by intricate correlations among learning rate, batch size, number of training tokens, and model size, as well as the prohibitive cost of hyperparameter optimization for large-scale models. Leveraging zero-shot transfer techniques, the authors explore dataset size as a fidelity parameter. They extend the WSD scheduler to focus on learning rate, batch size, and number of tokens, conducting extensive small-scale experiments that reveal a power-law relationship among these variables. This leads to the proposal of the Power scheduler, a learning rate scheduler agnostic to the number of training tokens and batch size. Key contributions include:

* Demonstrating that dataset size can effectively serve as a fidelity parameter for zero-shot transfer in hyperparameter tuning.
* Identifying a power-law relationship between learning rate, batch size, and number of training tokens.
* Proposing the Power scheduler to simplify hyperparameter tuning by reducing dependency on training tokens and batch size.

**Strengths:**

1. Strong Motivation and Practical Relevance: The paper addresses the complex challenge of optimizing learning rates in language model pretraining, where hyperparameters like learning rate, batch size, number of training tokens, and model size are deeply intertwined. Given that hyperparameter optimization at the target scale is computationally prohibitive, their effort to reduce the number of hyperparameters is both timely and valuable.

2. Overcoming Limitations of Existing Schedulers: The authors tackle the drawbacks of current learning rate schedulers, such as the cosine scheduler's requirement to specify the total number of training tokens—a constraint that limits its use in scenarios like continual learning. The proposed Power scheduler is agnostic to the number of training tokens and batch size, offering greater flexibility and applicability.

3. Empirical Discovery of a Scaling Law: Through extensive experiments, they demonstrate that the optimal learning rate shifts across different ranges of tokens and batch sizes, indicating it doesn't transfer directly across scales. By uncovering a power-law relationship among these variables, they provide an empirical foundation for their scheduler.

4. Extensive Experimentation and Practical Impact: The significant amount of experimentation not only validates their findings but also showcases the scheduler's effectiveness on 3B-parameter dense and MoE models. This contributes to simplifying hyperparameter tuning for large-scale language models, which has substantial practical implications for the field.

**Weaknesses:**

1. Limited Transferability Demonstrated: The paper's primary contribution relies on identifying specific coefficients for a power-law relationship among learning rate, batch size, and number of training tokens. However, the transferability of these coefficients to different models and datasets is not thoroughly explored. Except for the Mixture-of-Experts models based on the same language model used to uncover the power law, there is a lack of experimentation with other architectures or datasets. This gap leaves uncertainty about whether the proposed Power scheduler can be effectively applied to a broader range of models.

2. Insufficient Details on Hyperparameter Optimization: The methodology for determining the power-law coefficients lacks crucial details. It's unclear whether the authors employed grid search, random search, or another optimization strategy when tuning the coefficients $a$ and $b$ in Figure 5. Information about the number of configurations explored, the computational budget, and the overall search process is missing. This is particularly important if the coefficients are not universally transferable; practitioners aiming to apply the scheduler to different models would need guidance on how to find appropriate coefficients and understand the associated computational costs.

3. Missing Baseline Comparisons: The paper does not compare the proposed scheduler with simpler baselines, such as a warmup plus a stable learning rate scheduler. A basic scheduler that allocates a fixed percentage (e.g., $10$%) of training steps for warmup and then maintains a stable learning rate could also reduce hyperparameters and function well in continual learning setups. Including such baselines, possibly with a final cooldown period would strengthen the evaluation by providing a clearer context for the scheduler's performance improvements.

4. Absence of Learning Curves: Despite citing the Maximum Update Parameterization paper which focuses on stable training, the paper lacks learning curves (e.g., training loss versus number of tokens) that could illustrate training stability and dynamics. Including these curves would provide valuable insights into how the scheduler affects the training process over time and would support claims about improved stability or convergence rates.

5. Unclear Benefit Magnitude: While the experiments show that the optimal learning rate shifts across different token ranges and batch sizes, the practical significance of these shifts is not fully quantified. The performance improvements appear modest; for instance, in the experiments with the Dense model, the Power scheduler outperforms other methods in only 3 out of 7 tasks. A more detailed analysis of the benefits, including effect sizes and statistical significance, would help clarify the scheduler's impact.

6. Minor Typos and Missing Explanations: There are minor errors that need correction—Line 258 should use "T" instead of "x," and Figure 6 should label the coefficient as "b" instead of "a." Additionally, explanations for other models are missing and could be included in the appendix.

**Questions:**

1. Can you provide evidence of the power-law coefficients' transferability to other models and datasets?
2. What was your methodology and budget for determining the power-law coefficients $a$ and $b$?
3. What guidelines can you provide for practitioners on determining coefficients and required computational resources?
4. Have you compared the Power scheduler with simpler baselines like a (warmup plus) stable learning rate (+ final cooldown)?
5. Could you include learning curves to show training stability when using the Power scheduler?
6. How do you quantify the practical benefits of the Power scheduler given the modest improvements observed?
7. Have you tested the Power scheduler in continual learning scenarios with evolving dataset sizes?
8. Does the Power scheduler impact training efficiency regarding convergence speed or resource utilization?

---

### Official Review · Reviewer_y6ev · 2024-11-05

**Soundness:** 2
**Presentation:** 2
**Contribution:** 3
**Rating:** 3
**Confidence:** 5

**Summary:**

This paper proposes a new learning rate scheduler for language model pre-training that can accommodate varying numbers of pre-training examples. The authors identify a power-law relationship between the ratio LR / batch size and the number of training tokens. They then use this to design a scheduler that will decay the LR exponentially as training proceeds -- importantly, the scheduler improves over other options (cosine schedule) because it doesn't require knowing the number of tokens in advance.

Review summary: The problem is interesting, but there is a huge chunk of significant prior work (on scaling learning rates and batch sizes) that is missing. The plots of experiments appear to be incorrect, and most of the time, the takeaways do not match what is shown in the actual plot. Therefore, although the idea is interesting and may hold some merit, I am recommending rejection because the paper does not constitute a sound and valid scientific study.

**Strengths:**

1. It is interesting that the authors find that the learning rate of the WSD scheduler is dependent on the number of tokens trained. The motivation of the scheduler is that one should be able to train in the stable (S) phase for an arbitrary period of time and decay (D) whenever training is completed. However, I do see substantial issues with their experiments that make me question the validity of this finding.
2. Designing methods that transfer across training budgets is of great interest, since it's hard to ablate all crucial design choices at scale.

**Weaknesses:**

**Major Concerns**

1. The authors should cite theoretical and empirical work that has shown that the learning rate _does_ depend on the batch size [1,2,3]. Indeed, these prior works make Fig 2b and Hypothesis 1 unnecessary. Most of the points on their grid agree with the square root scaling rule introduced in [1] and theoretically justified in [2] -- I suspect that if they also adjusted the $\beta_1,\beta_2$ hyperparameters in Adam, then it would exactly agree with the square root scaling rule. The bigger problem is that this prior work implies there is _no way_ that the power scheduler can be agnostic to the batch size. Indeed, the main experiments simply pick a fixed batch size and thus don't substantiate the claim that this scheduler generalizes across batch sizes.
2. I am also concerned that there is not much mathematical thought put into any of the findings in the paper. Decisions are motivated heuristically, perhaps with incorrect plots of experiments, and this makes it hard to believe that the scheduler really will transfer robustly. It's not necessary to have formal proofs or anything like that in this paper, but I think it is necessary to provide intuition of some sort, either via math or careful analyses of the training curves.
3. The takeaway that I get from Fig 5 doesn't match the written takeaway. The plot shows that the choice of $a$ can drive changes of up to 10 points in the test perplexity, which is _extremely_ significant. Yet the caption and the text in lines 376-377 say that the test perplexity is not sensitive to the choice of $a$.
4. Altogether, given the issues with Figs 5 and 6, I think the values chosen for $a$ and $b$ don't make any sense to me. The authors should have plotted $(a,b)$ together against the test perplexity. As it stands, there's no evidence to drive the choice of $a$ and $b$.

**Minor Concerns**

1. My above point here also raises the question: why didn't the authors study the effect of $\beta_1,\beta_2$?
2. The paper makes several slightly incorrect statements about $\mu$P. For example, at the start of Section 3, the authors say that "Due to our use of $\mu$P, we expect the optimal learning rate to be invariant across different model sizes, batch sizes, and training steps." However, Tensor Programs (the theoretical framework behind $\mu$P) can only comment on the optimal learning rate on a fixed dataset. I suggest the authors look for these slight mis-statements throughout their paper and correct them. These culminate in the unsupported claim that the hyperparameters for their power scheduler will transfer across model sizes, training tokens, and batch sizes -- there's no fundamental reason for this to be true.
3. Fig 3 is not made correctly. For example, why is the red curve between batch size 32 and 64 exhibiting a non-zero second-order derivative? It should just be a line, unless additional batch sizes between 32 and 64 were tested. This is misleading. Actually, I am not even sure why these points are connected instead of being presented as a scatter plot.
5. Fig 6 is also confusing. The caption says it is about the $a$ hyperparameter but the plot appears to be for the $b$ hyperparameter, since the values are all negative.
7. It would be nice to see training curves for the 1B runs with different schedulers. That can help people understand what is going on when you use the Power Scheduler.
8. The significance of the 1B results is not apparent to me. The improvement is marginal, and there are more hyperparameters to tune with the Power Scheduler. Also, given the issues with Section 4, I can't tell if the hyperparameters are even selected correctly.

[1] You et al., *Large Batch Optimization for Deep Learning: Training BERT in 76 minutes* (ICLR 2020).

[2] Malladi et al., *On the SDEs and Scaling Rules for Adaptive Gradient Algorithms* (NeurIPS 2022).

[3] Goyal et al., *Accurate, Large Minibatch SGD: Training ImageNet in 1 Hour* (2017)

**Questions:**

See above.

---

### Note · Authors · 2024-12-19

I have read and agree with the venue's withdrawal policy on behalf of myself and my co-authors.